# Effect of Weight Thresholding on the Robustness of Real-World Complex Networks to Central Node Attacks

Jisha Mariyam John [1,*], Michele Bellingeri [2,3], Divya Sindhu Lekha [1], Davide Cassi [2,3] and Roberto Alfieri [2,3]

1 Indian Institute of Information Technology, Kottayam 686635, India; divyaslekha@iiitkottayam.ac.in
2 Dipartimento di Scienze Matematiche, Fisiche e Informatiche, Università di Parma, 43124 Parma, Italy; michele.bellingeri@unipr.it (M.B.); davide.cassi@unipr.it (D.C.); roberto.alfieri@unipr.it (R.A.)
3 Gruppo Collegato di Parma, National Institute of Nuclear Physics (INFN), 43124 Parma, Italy
* Correspondence: jishamariyam.phd201010@iiitkottayam.ac.in

**Abstract:** In this study, we investigate the effect of weight thresholding (WT) on the robustness of real-world complex networks. Here, we assess the robustness of networks after WT against various node attack strategies. We perform WT by removing a fixed fraction of weak links. The size of the largest connected component indicates the network's robustness. We find that real-world networks subjected to WT hold a robust connectivity structure to node attack even for higher WT values. In addition, we analyze the change in the top 30% of central nodes with WT and find a positive correlation in the ranking of central nodes for weighted node centralities. Differently, binary node centralities show a lower correlation when networks are subjected to WT. This result indicates that weighted node centralities are more stable indicators of node importance in real-world networks subjected to link sparsification.

**Keywords:** complex network; robustness; weight thresholding; node attack strategies; weak link removal





## 1. Introduction

The vulnerability of a complex network denotes the decrease in network functioning due to damage like the loss of some node or link. The ability of a network to withstand or overcome such situations is called its robustness. Over the last two decades, several studies have investigated the robustness of complex networks using several different attack strategies. Initial studies [1–3] on the vulnerability of complex networks are based on binary networks neglecting the intensity of the link/connection. However, real-world networks become more realistic when we consider the intensity of links, i.e., the link weights. For example, in a coauthorship network, the weight of the links connecting authors is the number of co-authored papers. For communication networks like the Internet, link weight can be the amount of data transferred.

The early research on network vulnerability focused on centrality-based node attack strategies, as critical nodes might be located at central positions in the networks. Widely used centrality-based node attacks are based on the node's degree and betweenness [2–5]. The weighted versions of these attack strategies are strength [6] and weighted betweenness [7], respectively. Other node attack strategies are based on different topological properties of the networks, like eigenvector centrality [5,8], closeness centrality [9,10], and clustering coefficient [5]. In addition, scientists investigated these attack strategies' variations [4,11,12]. By analyzing the impact of these attack strategies, we can identify the node importance in the network. Apart from these central attack strategies, influential nodes can be identified by various methods such as multiscale node importance measure [13] and classified neighbors algorithm [14].

The effect of the attack strategies is measured based on degradation in network performance (network functioning). The commonly used network functioning measures are the size of the largest connected component (LCC) [2–4,15], global efficiency [3,16], weighted efficiency [7,17], diameter [9], and total flow [15].

Apart from these node attack strategies, the robustness of complex networks has widely been studied in different link removal strategies. Removing a link can be interpreted in several real-world events, such as malfunctioning communication cable, damage of roads connecting two cities, prohibition of contact between individuals for controlling epidemic spreading, etc. Like node removal, the study of link removal started with link centrality concepts of betweenness and the degree of the link's end nodes [3,17,18]. The degree of a link can be defined as an aggregate (product, sum, minimum, and maximum) of the degree of end nodes of that link [3,18]. The betweenness of a link denotes the average number of shortest paths passing through it [3,17,18]. The weighted versions of these link centralities are defined in [17].

From Granovetter's "strength of weak ties hypothesis" in social networks [19], links are classified according to their weights as weak or strong. Several studies investigated the role of strong and weak links on network robustness. Pajevic and Plenz [20] found that the average clustering of nodes is robust against the removal of weak links but vulnerable when removing strong links. Link weight heterogeneity [15] is another factor that negatively affects the robustness of complex networks toward link removal.

The study of link removal in the economic complex system reveals that weak connections are more significant in supporting the overall connectivity of the system [21]. Group structures of complex networks are maintained even when most links are removed according to their increasing order of weight [22]. In addition, a widely recognized result is that "weak links are the universal key for complex network stability" [23]. These studies reveal the role of weak connections in maintaining functionality in real networks.

The high number of links makes time-expensive or cumbersome analyses on real-world networks. For this reason, scholars proposed different techniques for the sparsification of networks (i.e., to reduce link density) in these years [22,24]. Sparsification is a family of methods to build networks with a small number of links, often leading to a better generalization of the networks [25]. Sparse networks also have significantly lower computational costs than their denser counterparts, often two orders of magnitude in computational cost reduction [26]. This is especially relevant for the large and dense real-world and model networks that present prohibitively costly simulation analyses [27]. Thus, the sparsification methods applied to denser networks are helpful for reducing computational costs.

Weight thresholding (WT) is a sparsification approach to reduce link density in different real-world networks, such as financial, brain, and biological networks [28–30]. WT removes all links with a weight less than a particular threshold value. The objective of the WT procedure is to prune the highest number of links avoiding drastically altering the critical features of the original network. Unfortunately, many conventional network properties quickly change under the WT procedure [22,31].

Link shielding identifies critical links worth protecting [32–34]. As WT, link shielding techniques may be helpful to reduce the network links, thus improving computational feasibility by decreasing the computational cost. WT and link shielding procedures can be viewed as complementary methodologies for network sparsification.

This paper assesses how WT impacts the robustness measurement of weighted real-world networks when subjected to different node attack strategies. For example, let us consider the scenario of transportation systems. A transportation network is a network underlying the infrastructure that facilitates the movement of people/goods/services from one location to another. The demolition of one or more locations may affect the functionality of the entire infrastructure. The optimal planning of such a navigation system with multiple connections is computationally challenging. Nevertheless, this could be simplified by removing trivial connections (with less traffic). For this simplification, we can adopt weight thresholding (WT), in which a fixed fraction of weak links are removed

from the network. This link pruning improves the computational feasibility, but we are now apprehensive about the robustness of the thresholded network. Is the transportation network still robust as the initial network?

In this paper, we assess whether the weight thresholding alters the robustness of networks. We investigate this with a focus on robustness against node removal attacks. Our results highlight that real-world networks hold comparable robustness (in terms of LCC) to node attack strategies even after removing many weak links. In addition, we assessed how the ranking of central nodes changes with the weight thresholding procedure. We found that the node ranking remains positively correlated to the initial network when using weighted notions of node centralities.

## 2. Methods

### 2.1. Real-World Networks

We implemented five different attack strategies on nine real-world networks from different domains. The networks we used are weighted. The weight associated with the links depicts the empirical and specific characteristics of the networks. For example, in the case of the US airport network, the link weight indicates the number of passengers traveled per year [35]. In the coauthorship network Netscience, the link weight accounts for the number of papers co-authored between scientists [36]. We summarize the statistics of real-world networks in Table 1, with node, link, and link weight meaning. In addition, we furnish the reference for further information about each network.

**Table 1.** Statistics of real-world networks. N—number of nodes; L—number of links; <w>—average weight; <k>—average degree; LCC—size of the largest connected component.

| Networks | Key | Ref. | Type | Node | Link | Weight | N | L | <k> | <w> | LCC |
|---|---|---|---|---|---|---|---|---|---|---|---|
| *C. elegans* | Eleg | [37,38] | Biological | Neurons | Neurons connection | Number of Connections | 297 | 2344 | 15.8 | 3.761 | 297 |
| Cargoship | Cargo | [39] | Transport | Ports | Route | Shipping journeys | 834 | 4348 | 10.4 | 97.709 | 821 |
| US airport | Air | [35] | Transport | Airports | Route | Passengers | 500 | 2979 | 11.9 | 152,320.2 | 500 |
| *E. coli* | Coli | [39,40] | Biological | Metabolites | Common reaction | Number of Common reactions | 1100 | 3636 | 6.61 | 1.364 | 1100 |
| Netscience | Net | [36] | Social | authors | Coauthorship | Number of Common papers | 1461 | 2741 | 3.75 | 0.434 | 379 |
| Human12a | Hum | [41,42] | Biological | Brain regions | Connection between regions | Connection density | 501 | 6038 | 24.1 | 0.01 | 501 |
| Caribbean | Carib | [43,44] | Ecological Food web | Species | Trophic relation | Amount of biomass | 249 | 3503 | 28.13 | 0.067 | 249 |
| CypDry | Cyp | [45,46] | Ecological Food web | Species | Trophic relation | Amount of biomass | 66 | 503 | 15.24 | 0.358 | 65 |
| Budapest | Buda | [47] | Biological | Brain regions | Neural connection | Amount of track flow | 480 | 1000 | 4.167 | 5.024 | 467 |

### 2.2. Attack Strategies

We simulated network attacks by removing the nodes based on their centrality measures. The node centrality measures considered here include binary as well as weighted structure of the networks. The node attack strategies are:

- Random (*Ran*): Nodes are randomly removed. Random removal is analogous to errors or failures occurring in the network. Random failures are benchmark models in the study of network robustness [1,2].

- Degree (*Deg*): The degree of a node is a simple local centrality measure defined as the number of links connected to it. The degree $k_i$ of node $i$ is given by

$$k_i = \sum_{j=1}^{N} a_{ij},\tag{1}$$

where $a_{ij} = 1$ indicates the presence of a link between nodes $i$ and $j$ and is 0 otherwise. $N$ is the number of nodes in the network. The degree attack strategy first removes nodes with the highest degree (hubs). Earlier studies of network robustness to targeted attacks are based on this strategy [1,3–5,48].

- Strength (*Str*): A node's strength is the sum of the weights of links connected to that node. It is a weighted version of the degree centrality [6].

  Mathematically, the strength $s_i$ of node $i$ is:

$$s_i = \sum_{j=1}^{N} a_{ij} \cdot w_{ij},\tag{2}$$

where $a_{ij} = 1$ indicates the presence of a link between nodes $i$ and $j$ and is 0 otherwise. $w_{ij}$ is the weight of the link between $i$ and $j$. In this attack strategy, nodes with the highest strength are removed first.

- Betweenness (*Bet*): Betweenness of a node is the number of shortest paths (between all the pairs of nodes) passing through it [3–5]. This binary metric defines the shortest path between two nodes as the minimum number of links needed to travel from one node to another. Mathematically, betweenness $b_i$ of node $i$ is:

$$b_i = \sum_{s,t=1}^{N} \frac{\sigma_{st}(i)}{\sigma_{st}}\tag{3}$$

where $\sigma_{st}(i)$ is the number of shortest paths between nodes $s$ and $t$ passing through the node $i$. $\sigma_{st}$ is the total number of shortest paths between nodes $s$ and $t$. Based on this global metric, attack strategies remove nodes with the highest betweenness first.

- Weighted betweenness (*WBet*): Weighted betweenness of a node is defined as the number of weighted shortest paths passing through that node [7].

  Weighted betweenness $b_i^w$ of node $i$ is:

$$b_i^w = \sum_{s,t=1}^{N} \frac{\sigma_{st}^w(i)}{\sigma_{st}^w},\tag{4}$$

where $\sigma_{st}^w(i)$ is the number of weighted shortest paths between nodes $s$ and $t$ passing through the node $i$. $\sigma_{st}^w$ is the total number of weighted shortest paths between nodes $s$ and $t$.

While computing betweenness, it is essential to differentiate whether the link weight corresponds to "flows" or "costs" [49]. If link weight means flow, such as the number of passengers in transportation networks or the number of common papers in authorship networks, then the shortest path is computed by summing the inverse of link weights. If link weights are costs such as distance or time of information delivery between two stations, shortest paths are computed directly by summing the link weights.

These attacks are performed by removing all nodes and the links incident on them. We performed initial (not recalculated) and recalculated (also named adaptive) attack strategies for each node centrality. The term initial attack means we compute the node rank on the initial network and remove the nodes in that order. Here, node ranks are not updated during the node removal process [3]. On the other hand, in recalculated attack strategies, node centrality values are recalculated after the removal of each node [3]. In the case of ties (i.e., nodes with equal centrality value), we randomly select the node to remove. These node ties are randomized by averaging the outcomes over 100 simulations.

### 2.3. Network Robustness Indicator

The largest connected component (LCC) is defined as the number of nodes in the giant component of the network, i.e., the largest number of connected nodes [1,2,48]. It is a commonly used binary measure for network robustness. It only gives a topological description of the networks. Here, normalized LCC (on initial LCC value), as a function of the fraction (q) of removal of nodes, is used as the measure for network damage. Normalized LCC allows the comparison of robustness across different networks. The attack strategies terminate when the network becomes wholly destructed (LCC becomes 1).

To compare the response of the networks to each attack strategy, we used the robustness R [16]. It is a single number [15] indicating the area under the curve of the network functioning against a fraction of nodes or links removed. Here, LCC is used as a network functioning indicator. The theoretical range of R is from 0 to 0.5. For example, Figure 1 left chart shows the LCC plot as a function of fraction q of removals for five node attack strategies (initial attack) on the *C. elegans* network. The right chart in Figure 1 reports the robustness outcome R of each attack strategy computed by the area under the LCC curve.

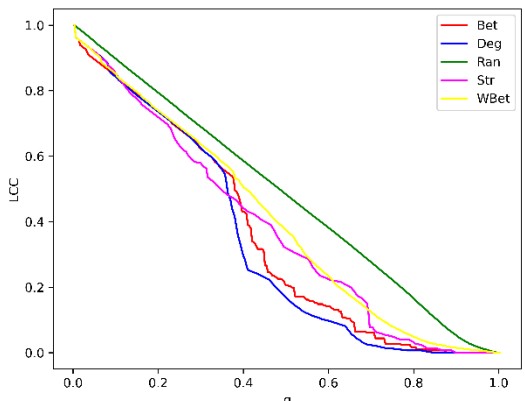 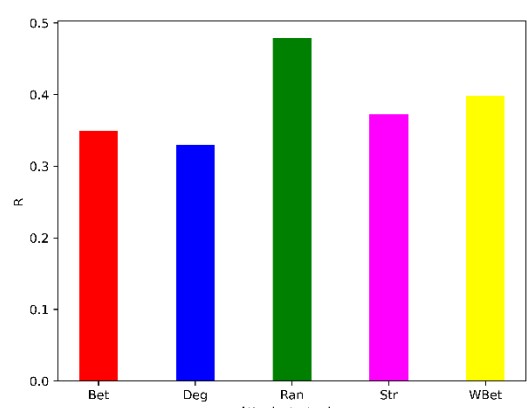

**Figure 1.** Left chart: LCC size as a function of the fraction of nodes removed q for initial attacks in *C. elegans* network. Right chart: Robustness R of each attack strategy.

### 2.4. Weight Thresholding

We investigated the effect of weak link removal on the robustness of real-world networks under various node attack strategies. This analysis was performed by the weight thresholding (WT) technique. Given a weighted network G with N number of nodes, and L number of links, the first step is to rank the links in increasing order of weight. The links of lower weight are considered weak links. Then, we performed the WT by removing a fraction of the weak links. For example, for WT = 0.05, we removed the first 5% weaker links in the rank. Consider a network with ten links of the following discrete weights: 1, 1, 2, 2, 4, 6, 7, 8, 8, and 9. Then, by WT = 0.5, we remove the links of weights 1, 1, 2, 2, and 4, in that order.

In our study, we took nineteen discrete threshold values WT = {0.0, 0.05, 0.1, 0.15, 0.2, 0.25, 0.3, 0.35, 0.4, 0.45, 0.5, 0.55, 0.6, 0.65, 0.7, 0.75, 0.8, 0.85, 0.9} (i.e., from 0% to 90% of weak links removal). In the case of ties (links with the same weight), we selected the links randomly. These ties are randomized by averaging the outcomes over 100 simulations. The thresholded network G′ will be the subgraph of G with the same number of nodes N and number of links, L′ = (1 − WT) L. Then, the node attack strategies on G′ are applied by identifying the nodes in the decreasing order of their centrality measures (*Deg*, *Bet*, *Str*, and *WBet*) computed from G′. This procedure is repeated for each WT. The overall methodology is depicted in Algorithm 1. The variables *m* and *n* in Algorithm 1 represent the number of iterations to break the link and node ties.

---

**Algorithm 1:** Methodology of WT analysis.

---
    Procedure Weight Thresholding (G, N, L)
1:    WT = {0.0, 0.05, 0.1, . . . . . . . . . . ., 0.85, 0.9}
2:    for each WT
3:        for $i = 1$ to $m$
4:            link_set = {links in the increasing order of their weight}
5:            weak_linkset = {WT fraction of weak links from link_set}
6:            G′ = G − weak_linkset
7:            Initial attack (G′, N, L′)
8:            Recalculated attack (G′, N, L′)
    Procedure Initial attack (G′, N, L′)
1:    Find Initial LCC
2:    for $i = 1$ to $n$
3:        node_set = {nodes of G′ in the decreasing order of centrality measure}
4:        while (LCC ! = 1)
5:            Remove a node x from the G′ (in the order of node_set)
6:            Find LCC of new network
7:            node_set = node_set − x
    Procedure Recalculated attack (G′, N, L′)
1:    Find Initial LCC
2:    for $i = 1$ to $n$
3:        while (LCC ! = 1)
4:            Calculate centrality meaures
5:            node_set = {nodes of G′ in the decreasing order of centrality measure}
6:            Remove a node x from the G′ (in the order of node_set)
7:            Find LCC of new network
8:            node_set = node_set − x

---

## 3. Results and Discussion

Removal of an entity of a network (either node or link) may result in changes in the network functionality after a particular fraction of removals. However, an entity is significant if its removal triggers a rapid decrease in the network functioning measure. For example, the black curve in Figure 2 indicates a sharp decrease in the network functioning along with the removal process. In contrast, gentle changes in the blue curve indicate the network's ability to withstand comparable functionality. The ability of a network to continue with comparable functionality can be an indicator of the network's robustness.

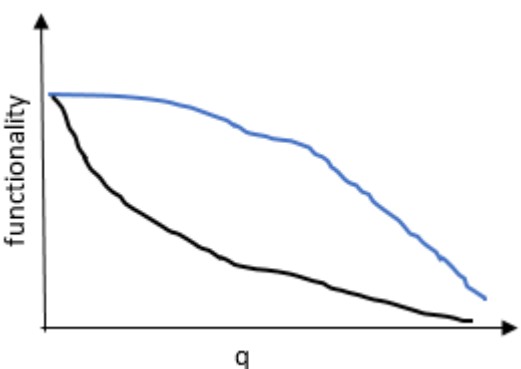

**Figure 2.** Steeper (black) and gentle (blue) degradation of a network functionality along the removal of fraction (q) of components (either nodes or links) of the network.

Here, we investigate the role of weak links in the robustness of networks to different node attack strategies. The analysis applied the WT procedure to the nine real networks. We simulated five node attack strategies, such as *Ran*, *Deg*, *Str*, *Bet*, and *WBet*, on these thresholded networks. For each strategy, we performed both initial and recalculated attacks. The WT procedure is performed by removing a fixed fraction of weak links. Figures 3 and 4 show the LCC and robustness (R) as a function of WT value for different node attack strategies and each real-world network.

The networks *C. elegans*, Caribbean, and Human12a show the slowest LCC decrease when subjected to the WT procedure. Specifically, *C. elegans* and the Caribbean have almost the same LCC after each thresholding even up to WT = 0.60, and Human12a does not show any degradation in LCC for WT ≤ 0.55.

The smallest network in our study, Cypdry, also (N = 66) maintains comparable LCC up to WT = 0.45. The other networks, such as *E. coli*, Budapest, Cargoship, and US Airports, present low robustness against WT procedure, showing a faster LCC decrease than other networks. In particular, US Airports and Budapest networks show faster LCC disruption under the WT procedure.

In summary, except for Budapest and US Airports networks, the real-world networks under study are robust to the WT procedure. The WT procedure corresponds to weak link removal [15,17]; for this reason, the real-world networks under study unveil general robustness to weak link removal.

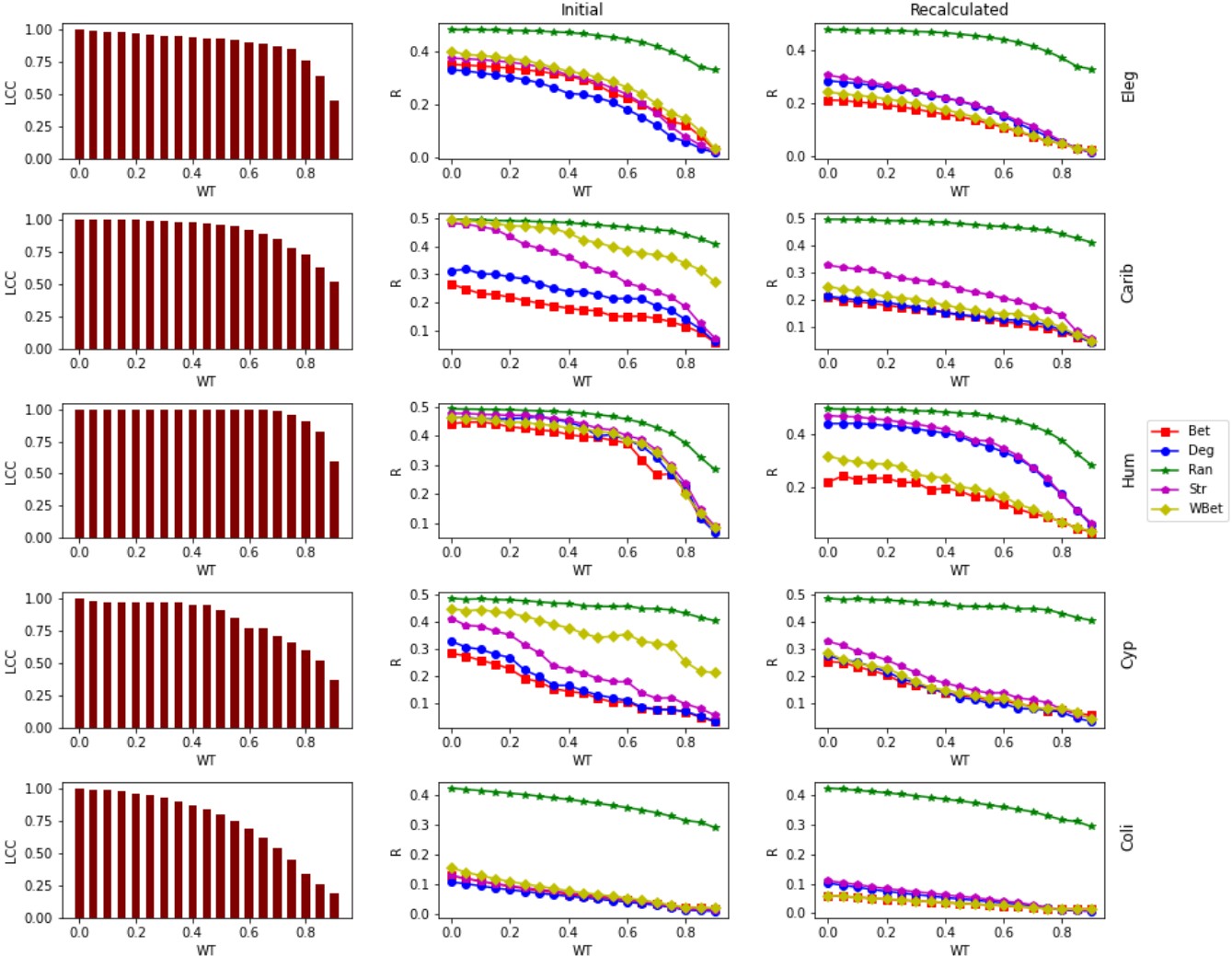

**Figure 3.** LCC after each weight thresholding (WT) value (**left column**), robustness (R) of the network under initial (**middle column**), and recalculated attack strategies (**right column**) as a function of weight thresholding (WT) value for the networks *C. elegans* (Eleg), Caribbean (Carib), Human12a (Hum), Cypdry (Cyp), and *E. coli* (Coli).

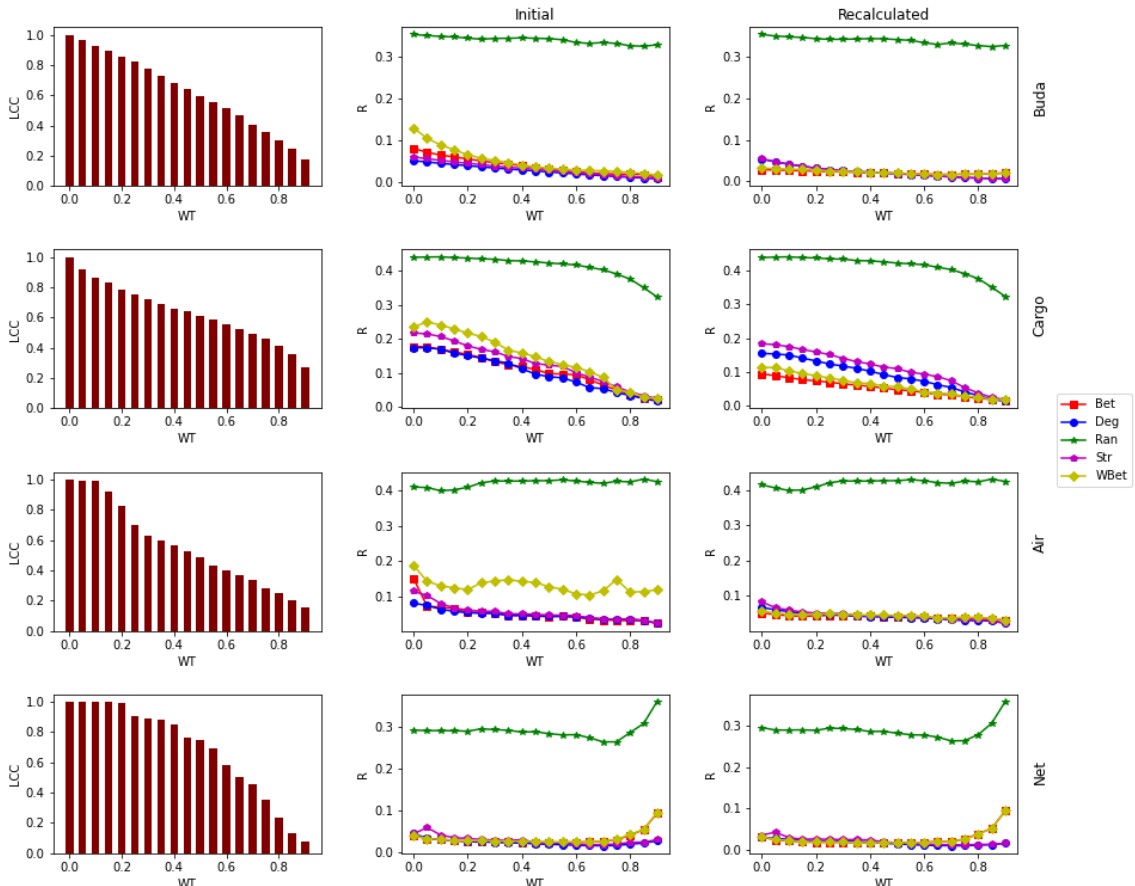

**Figure 4.** LCC after each weight thresholding (WT) value (**left column**), robustness (R) of the network under initial (**middle column**), and recalculated attack (**right column**) strategies as a function of weight thresholding (WT) value for the networks Budapest (Buda), Cargoship (Cargo), US Airports (Air), and Netscience (Net).

We can see the robustness (R) of different node attack strategies as a function of WT in Figures 3 and 4. R generally does not show a steeper decrease with WT for most node attack strategies, both initial and recalculated attacks. The transformation of robustness from the original network to the thresholded network after 90% removal of the weak link is evident but gradual. Therefore, in the real-world networks under study, we find general robustness against node attacks when subjected to the WT procedure. While increasing the WT value, we observe a very slight decrease in the robustness R to random node removal (*Ran*) (Figures 3 and 4, green curves). This result indicates that networks maintain the well-known "error resistance" feature [2] even when subjected to the WT procedure.

The gradual change in the robustness of each thresholded network to various node attack strategies furnishes interesting insights. On the one hand, it may indicate that the remaining network shows a robust connectivity structure to node attacks. Since the WT procedure decreases the LCC, we can argue that the remaining LCC is robust to node attack. On the other hand, the WT procedure does not cause a node rank change toward a more harmful node attack sequence. This last result indicates that the node centralities ranking is stable to the WT procedure.

There are exceptions. In Cypdry and Caribbean networks, the robustness R of the *Str* strategy decreases faster than other strategies (Figure 3, purple line). *Str* removes nodes according to their strength, i.e., the sum of the link weights of that node [6]. Further, we observe a similar decrease in network robustness R for the *WBet* (Figure 3, yellow line) strategy that removes nodes according to their weighted betweenness [7]. Therefore, [46] the WT procedure enhances the efficacy of the *Str* and *WBet* node attack to dismantle food

webs. The higher efficacy of *Str* and *WBet* can be due to a change in node ranking for these strategies tuning different WT values, with more effective node ranking when increasing WT. Food webs are ecological networks describing "who eats whom" in ecosystems, i.e., in these networks, nodes are biological species, and links depict trophic interactions among them [46,50]. These results suggest that removing weak links in food web ecological networks may unveil essential nodes in these ecological networks.

In Netscience, we can observe a rise in the robustness towards the end of the thresholding for betweenness-based attack strategies (*Bet* and *WBet*). Betweenness-based attack strategies show low efficacy when tuning higher WT. The LCC of the Netscience is only 24.9% of the overall size of the network (see Table 1), and the network contains a large number of components C (at WT = 0, C = 268, and at WT = 0.9, C is 1211). The LCC contains many nodes with low betweenness centrality values, attack strategies remove nodes from other components, and LCC remains unchanged when removing nodes according to their betweenness. This result indicates the necessity of conditional betweenness attack strategies [51].

The results found in other studies, such as recalculated attack strategies are more efficient than initial attack strategies, are also confirmed in our results. In initial attacks, binary strategies outperform weighted attacks. In recalculated attack strategy, *Bet* and their weighted version, *WBet*, are more efficient than *Deg* and *Str* for destroying LCC. With the increase in the fraction of weak link removal, the efficiency of attack strategies becomes closer. It indicates that the weighted structure has less significance in thresholded networks compared to original networks. In addition, all the networks are robust to random attacks ($R \approx 0.5$).

*Analyzing Node Centrality Ranking under WT by Kendall's Tau Coefficient*

Kendall's tau coefficient ($\tau$) is used to analyze the change in node rank after weight thresholding [52]. It is a measure of the degree of correspondence between two ranked data. Kendall's tau coefficient between two arrays of ranking A and B is

$$\tau = \frac{(n_p - n_q)}{\text{sqrt}((n_p + n_q + n_t) * (n_p + n_q + n_u))}, \tag{5}$$

where $n_p$ and $n_q$ are the numbers of concordant and discordant pairs, respectively; $n_t$ is the number of ties only in A; and $n_u$ is the number of ties only in B. If a tie occurs for the same pair in both A and B, it is not added to $n_t$ or $n_u$. The higher Kendall's tau coefficient, the more similar the two ranking sequences. The range of Kendall's tau coefficient is from $-1$ to 1.

This paper analyzed the correlation between the centrality ranking of the initial network's top 30% central nodes with each thresholded network (See Figure 5). We measured the correlation for four centralities *Deg*, *Str*, *Bet*, and *WBet*. When we compare the correlation of different centrality measures among all the networks along WT values, *Str* (purple line) is the more stable node ranking, followed by *WBet*. On the contrary, *Bet* (red line) and *Deg* (blue line) show higher variance in the centrality measure. Therefore, when subjected to the WT procedure, the weighted node centrality rankings (*Str* and *WBet*) are more stable than the binary counterparts (*Deg* and *Bet*). For example, the networks Caribbean, Human12a, Cypdry, Cargoship, and US Airports hold a correlation for weighted node centralities approximately above 0.4.

The *Deg* of Netscience shows a deep variation for initial WTs up to 0.3. This is because the number of connected components in the Netscience is high, and the top 30% of *Deg* central nodes are distributed among various components. Nonetheless, the other node centralities ranking are more stable to the WT procedure.

When taking these results together, we can point out that node centralities based on weighted features of the network show a more stable node ranking with the WT procedure.

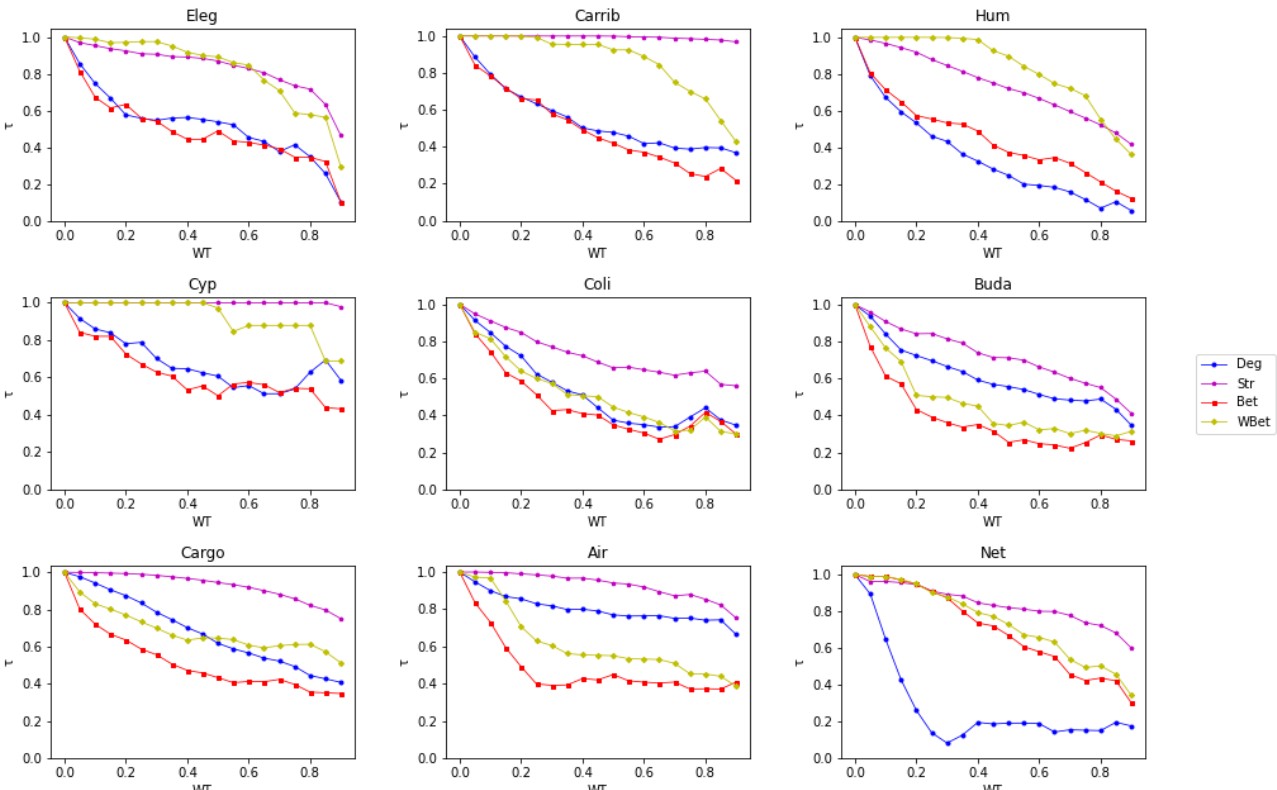

**Figure 5.** Kendall's tau coefficient ($\tau$) for centrality measures *Deg*, *Str*, *Bet*, and *WBet*. Correlation is measured between the top 30% of nodes of the initial network with each thresholded network.

## 4. Conclusions

We performed weight thresholding on real-world weighted networks. Here, weight thresholding corresponds to the removal of weak links. We analyzed the WT impact on the network's robustness to node attack strategies in initial and recalculated scenarios. In general, networks maintain their robustness structure regarding LCC along the WT procedure. In other words, weak link removal does not impact the LCC of the network, and the resulting thresholded networks show robust connectivity structures against node attacks. In addition to this, weighted node centralities hold a positive correlation with the ranking of most central nodes in the networks for different WT values. Differently, binary node centralities show low correlation when networks are subjected to WT.

With this result, weak link removal can be used as a method for the sparsification of the networks in which robustness to node attack is crucial.

Another interesting network sparsification approach is "link shielding" (LS), which is a method of identifying critical links worth protecting [32–34]. The weight thresholding investigated here is a complementary approach of LS for network sparsification. WT removes links under a certain weight threshold, whereas LS holds important links for the network. Both techniques improve computational feasibility by reducing simulation costs. For this reason, it would be very interesting to analyze the robustness against node removal of networks subjected to LS and compare the outcomes with the results presented in this research.

Lastly, adopting LCC as a measure of the network is one-sided. Therefore, as a follow-up to this work, we can extend our study with other robustness indicators, such as efficiency. Also, we can extend the study by analyzing the impact of strong link removal on the network's robustness to various node attack strategies.

**Author Contributions:** Conceptualization, M.B. and D.C.; methodology, J.M.J. and M.B.; formal analysis, J.M.J.; investigation, J.M.J. and D.S.L.; writing—original draft, J.M.J. and M.B.; writing—review & editing, M.B., D.S.L., D.C. and R.A.; visualization, J.M.J.; supervision, D.S.L. All authors have read and agreed to the published version of the manuscript.

**Funding:** This research is funded by the IIT Palakkad Technology IHub Foundation Doctoral Fellowship IPTIF/HRD/DF/019 and Ecosister project, funded under the National Recovery and Resilience Plan (NRRP), Mission 4 Component 2 Investment 1.5—Call for tender No. 3277 of 30 December 2021 of Italian Ministry of University and Research funded by the European Union—NextGenerationEU [2]Award Number: Project code ECS00000033, Concession Decree No. 1052 of 23 June 2022 adopted by the Italian Ministry.

**Data Availability Statement:** All required data are provided in the manuscript.

**Conflicts of Interest:** The authors declare no conflict of interest.

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
