# Peer review of "Effect of Weight Thresholding on the Robustness of Real-World Complex Networks to Central Node Attacks"

_mathematics, doi:10.3390/math11163482_

Round 1

Reviewer 1 Report

In this manuscript, the authors explored the robustness of networks after weight thresholding (WT) is introduced to various node aÄ´ack strategies, where WT is performed by removing a fixed fraction of weak links. In addition, the network’s robustness is indicated by the size of the largest connected component. Through extensive numerical simulations, they found that real-world networks subjected to WT hold a robust connectivity structure to node aÄ´ack even for higher WT values.

In my opinion, the idea could be interesting and the results could be believable, but I have the following concerns

1) What is the weight thresholding? Why or How do the authors perform this kind of operations? That is, the motivation is not enough.

2) In the WT, the authors completed the WT by removing a fixed fraction of weak links.  What are the weak links? Why or how do the authors indentify the weak links?

3) Some typos exist. For examples, in Tab.1, the table should be drawn in the three-line style; After Eqs (1)-(4), there should be a comma after equation.

4) Regarding the robustness, I suggest the authors make some comparisions in two class complex network models, Barabasi-Albert scale-free and Watts-Strogatz small world networks among different attack strategies.

5) The references are incomplete. Reagrding the indicatiors characterzing the node importance, two related works need to be mentioned here. 

1) Identification of influential spreaders based on classified neighbors in real-world complex networks. Applied Mathematics and Computation, 2018, 320, 512-523.
2) Node importance for dynamical process on networks: A multiscale characterization. Chaos: an interdisciplinary journal of nonlinear science, 2011, 21(1), 016107.

The english is poor, and I wish the authors make a thorough check. such as

1) In Abstract, 

"we assess the robustness of networks after WT to various node 10
aÄ´ack strategies"-->"we assess the robustness of networks after WT is introduce to various node Ä´ack strategies" could be better.

2) In Line 67,  "cumbersome different analyses"-->"cumbersome analyses"

3) There should be a comma after each equation;

4) All references have the same style, such as [34] have a redundant dot.

Reviewer 2 Report

The paper studies on network robustness of complex networks under weight threshold (WT) based link removal followed by intentional node attack. It is concluded that WT based link removal does not significantly change network robustness measured by LCC. The paper appears to be clearly written, making it generally speaking not difficult to follow. The only issue is that I am not really convinced that the main conclusion of the paper is of significant value.

If my understanding is correct, weights of the links are randomly assigned, making WT-based link removal essentially a random link removal process. If such an understanding is correct, it may be relatively trivial to point out than having random link removal before node attack generally speaking does not significantly affect the final LCC. Specifically, it is well understood that (1) random link removal does not easily split a network into disconnected big components though a small number of nodes, especially some low-degree ones, may become isolated; (2) random link removal does not significantly change the rank of nodal degrees of different nodes. Combining the above two points, we could see that having WT-based link removal first followed by node attack may perform rather similarly as that of the case where we have node attack first and then random link removal. The random link removal does not significantly reduce the size of LCC. Thus it may not be a surprise that LCC is not significantly reduced.

In fact, the above conclusion probably can be mathematically analyzed by some relatively straightforward extensions of Barabasi and Havlin’s theoretical analyses on network robustness in year 2000 and year 2001.  

I may have missed or misunderstood something that is very important. But if my above understanding is correct, I find it difficult to recommend acceptance of the paper in its current form.

Reviewer 3 Report

The authors investigate the effect of weight thresholding on the robustness of real-world complex networks to various node attack strategies. They apply the weight thresholding in order to remove weak links. They perform well-conducted experiments and conclude that the weak link removal can be used as a method for the sparsification of the networks in which robustness to node attack is crucial.

I believe that they paper is technically sound and can be accepted for publication.

Reviewer 4 Report

In my opinion, it is a well-worked article, it is clear in its presentation and in the objectives that the authors set out and in their way of developing it. Except for a slightly more careful reading of the English, the article does not generate any major comments.

A slightly more careful reading of the English is suggested.

Round 2

Reviewer 1 Report

The authors have made the necessary revisions, and I am glad to recommend it to be accepted in the present form.

Author Response

We are delighted with the Reviewer's recommendation for acceptance.

Reviewer 2 Report

I have to say that I am disappointed to see the authors' surprisingly simply answer to my former comments. If the link weight is NOT randomly assigned, how is it assigned? Is it decided by the degrees of the end nodes of the link, the centrality? the flow going through the link? or something else? Simply saying that the links carry weight in real-life networks does NOT answer my question as the link weight in the real-life networks still has to be defined in a certain way. What is that definition that has been adopted in the paper? Without getting a clear answer to my main concern, I am afraid that I cannot change my recommendation as it is not clear to me that the results are of significant merits.

Author Response

Thanks to the Reviewer for this suggestion. Since we are studying real-world networks, we are not assigning link weights as part of our experiment. The weights associated to the links in these networks are empirically collected by the different authors building the networks. Therefore, the weights associated to the links depict specific and empirical characteristics of the respective networks. For example, in the case of the US airport network, the weight of each link indicates the number of passengers travelled per year. In the co-authorship network Netscience, the weights of each link are assigned by the number of papers co-authored among scientists. Table.1, under the "Methods" section, describes the source reference of each network (Col. ‘Ref.’), and what the node, link, and weight represent in each network.

Following the Reviewer's suggestion, we added a part in the Method section explaining the empirical nature of the link weight in our real-world networks dataset (Row 106, blue part):

“The networks we used are weighted. The weight associated with the links depicts empirical and specific characteristics of the networks. For example, in the case of the US airport network, the link weight indicates the number of passengers traveled per year [32]. In the co-authorship network Netscience, link weight accounts for the number of papers co-authored between scientists [33].  We summarize the statistics of real-world networks in Table 1, with node, link, and link weight meaning. In addition, we furnish the reference for further information about each network.”

Round 3

Reviewer 2 Report

I deeply appreciate the authors' patient responses to my last-round comments. Now I understand that the link weightage may denote different things in different systems. With such understanding, however, my feeling is still that the main results of the paper are probably not a surprise. As mentioned in earlier-round reviews, the reported results can be related to some existing studies on network robustness, in different names (e.g. effects of link shielding [1,2] and imperfect node protection [3] etc.)  Having said that, I have to admit that I have not seen a reference talking about the effects of exactly the same link removal method. For that, the paper may be worth publication after all. I would like to suggest that the authors may give a better survey of some related existing results and clearly explain how the results reported in this paper are different from these results, and why the new results are so valuable.  

[1] Zhang et al, Enhancing network robustness via shielding, IEEE/ACM Trans. Network., 2017.

[2]  Liu et al., Optimizing communication network geodiversity for disaster resilience through shielding approach, Reliab. Engin. Syst. Safety, 2022.

[3] Xiao et al., On imperfect node protection in complex communication networks, J. Physics A, 2011.

Author Response

We are thankful to the Reviewer for the fruitful suggestion. The Reviewer furnishes good references suggestion for our manuscript. We added these references by differentiating link shielding and weight thresholding in the Introduction and Conclusion part.

Introduction:

R84-87: Link shielding identifies critical links worth protecting [32, 33, 34]. As WT, link shielding techniques may be helpful to reduce the network links, thus improving computational feasibility by decreasing the computational cost. WT and link shielding procedure can be viewed as complementary methodologies for network sparsification.

Conclusion:

R342-348: Another interesting network sparsification approach is ‘link shielding’ (LS), which is a method of identifying critical links worth protecting [32, 33, 34]. The weight thresholding investigated here is a complementary approach of LS for networks sparsification. WT removes links under a certain weight threshold, whereas LS holds important links for the network. Both techniques improve computational feasibility by reducing simulation cost. For this reason, it would be very interesting to analyze the robustness against node removal of networks subjected to LS and compare the outcomes with the results presented in this research.